# Metabolomic Insights into COVID-19 Severity: A Scoping Review

**DOI:** 10.3390/metabo14110617

**Published:** 2024-11-12

**Authors:** Eric Pimentel, Mohammad Mehdi Banoei, Jasnoor Kaur, Chel Hee Lee, Brent W. Winston

**Affiliations:** 1Department of Critical Care, Cumming School of Medicine, University of Calgary, Calgary, AB T2N 4Z6, Canada; eric.pimentel@ucalgary.ca (E.P.); mmbanoei@ucalgary.ca (M.M.B.); jasnoor.kaur@ucalgary.ca (J.K.); chelhee.lee@ucalgary.ca (C.H.L.); 2Department of Biological Sciences, University of Calgary, Calgary, AB T2N 4Z6, Canada; 3Department of Mathematics and Statistics, Faculty of Science, University of Calgary, Calgary, AB T2N 5A1, Canada; 4Departments of Medicine, Biochemistry and Molecular Biology, Cumming School of Medicine, University of Calgary, Calgary, AB T2N 4Z6, Canada

**Keywords:** metabolomics, COVID-19, SARS-CoV-2, disease severity, acute respiratory distress syndrome (ARDS), scoping review

## Abstract

Background: In 2019, SARS-CoV-2, the novel coronavirus, entered the world scene, presenting a global health crisis with a broad spectrum of clinical manifestations. Recognizing the significance of metabolomics as the omics closest to symptomatology, it has become a useful tool for predicting clinical outcomes. Several metabolomic studies have indicated variations in the metabolome corresponding to different disease severities, highlighting the potential of metabolomics to unravel crucial insights into the pathophysiology of SARS-CoV-2 infection. Methods: The PRISMA guidelines were followed for this scoping review. Three major scientific databases were searched: PubMed, the Directory of Open Access Journals (DOAJ), and BioMed Central, from 2020 to 2024. Initially, 2938 articles were identified and vetted with specific inclusion and exclusion criteria. Of these, 42 articles were retrieved for analysis and summary. Results: Metabolites were identified that were repeatedly noted to change with COVID-19 and its severity. Phenylalanine, glucose, and glutamic acid increased with severity, while tryptophan, proline, and glutamine decreased, highlighting their association with COVID-19 severity. Additionally, pathway analysis revealed that phenylalanine, tyrosine and tryptophan biosynthesis, and arginine biosynthesis were the most significantly impacted pathways in COVID-19 severity. Conclusions: COVID-19 severity is intricately linked to significant metabolic alterations that span amino acid metabolism, energy production, immune response modulation, and redox balance.

## 1. Introduction

SARS-CoV-2 is an RNA virus of the Coronaviridae family. The causative agent was responsible for Coronavirus Disease 2019, and it targeted predominantly the respiratory tract. The viral spike protein contains a receptor-binding domain, which binds with a high affinity to the angiotensin-converting enzyme 2 (ACE2) receptor that is highly expressed on the surface of epithelial cells in the respiratory tract and lungs, intestines, heart, and other tissues [1]. The binding of the viral receptor-binding domain is via the S1 subunit. After binding, the spike protein undergoes a proteolytic (activation) step by a host cell protease, the transmembrane serine protease 2 (TMPRSS2). The activation enables the fusion of the viral envelope with the host cell membrane. The fusion of the viral and cellular membranes is mediated through the S2 subunit of the spike protein [2]. Following the fusion of both membranes, the viral RNA genome is released into the cytoplasm of the host cell. The resulting internalized viral RNA then serves as a template for viral replication and transcription, producing new viral particles, thus initiating repetitive infection.

As of 2024, there have been over 775 million confirmed COVID-19 cases reported worldwide [3]. The cumulative global figures for total hospitalizations and ICU admissions throughout the COVID-19 pandemic are challenging to pinpoint due to the variability in the data reporting across countries. However, it is estimated that over a 28-day period between 24 June and 21 July 2024 there have been over 23,000 new hospitalizations and more than 600 new ICU admissions globally [4]. All these differences between the confirmed cases, hospitalizations, and admissions to the ICU not only reflect the ongoing burden that COVID-19 poses to healthcare, but also the varying degrees of severity in COVID-19 cases. Indeed, SARS-CoV-2 does cause a spectrum of clinical manifestations, ranging from mild illness to respiratory failure with acute respiratory distress syndrome, which is the most severe form of the disease, necessitating ICU admission.

According to the National Institutes of Health, NIH [5], COVID-19 severity is defined as:Mild illness: patients with variable symptoms related to COVID-19—a general feeling of being unwell, headache, fever, muscle pain, sore throat, cough, rhinorrhea—but no shortness of breath, dyspnea, or abnormal chest imaging.Moderate illness: patients with lower respiratory disease, as evidenced by history, exam findings, or imaging, and have oxygen saturation measured by pulse oximetry (SpO_2_) ≥ 94% on room air at sea level.Severe illness: a patient with a SpO_2_ < 94% on room air at sea level, a ratio of arterial partial pressure of oxygen to fraction of inspired oxygen (PaO_2_/FiO_2_) < 300 mm Hg, a respiratory rate > 30 breaths/min, or lung infiltrates of >50%.Critical illness: anyone suffering from respiratory failure, septic shock, or multiple organ dysfunction.It is, therefore, essential to have tools that would enable an ability to recognize, and possibly prevent, the progression of COVID-19 severity by helping clinicians with treatment adjustments to avoid further progression and complications.

In effect, SARS-CoV-2 triggers a metabolic reprogramming of host cells to deliver an environment that supports its replication and survival. This type of reprogramming ultimately impacts a series of metabolic pathways related to amino acid, glucose, and lipid metabolism [6].

Metabolomics studies have successfully identified several metabolic signatures of various diseases, including cardiovascular diseases, neurodegenerative diseases, and metabolic disorders [7]. Research has now advanced from identifying single markers of disease to the linkage of comprehensive blood metabolomic profiles with aging [8], the onset of diseases, disease severity, and markers associated with mortality [9], thus highlighting the human blood metabolome as a true mirror of the body’s physiological state and of a disease state.

For COVID-19 infection, metabolomics studies have been carried out to characterize the difference between COVID-19 and influenza infections, in order to characterize differences between the different waves of COVID-19 [10] and to assess the severity of infection [11]. At the same time, the effects of certain treatments can and have been assessed. Not surprisingly, metabolites may even yield clues about the host’s immune status towards COVID-19 infection [12]. Evidence has also shown that a person’s metabolism may remain dysregulated after acute infection and persist, resulting in a presentation termed “long-COVID” [13].

In this scoping review, we will summarize the current body of metabolomics research to understand the serum or plasma metabolic changes implicated in acute COVID-19, with a particular emphasis on lung involvement and disease severity.

## 2. Materials and Methods

This scoping review has been carried out in conformity with Preferred Reporting Items for Systematic Reviews and Meta-Analyses (PRISMA) guidelines [14]. The principal databases consulted were PubMed, the Directory of Open Access Journals (DOAJ), and BioMed Central in an attempt to find comprehensive material within this scope of study.

### 2.1. Search Strategy

The objective of this scoping review was to systematically identify and evaluate all publications between 2020 and 2024 inquiring about metabolomics in COVID-19 severity and progression and present these data as a scoping review. This means accounting for specific metabolites and pathways that affect the course of COVID-19. In this regard, a targeted literature search strategy was implemented using exact keywords and phrases such as: “metabolomics” OR “metabonomics”, AND “COVID-19”, AND/OR “severity”, OR “ARDS”.

The search across databases yielded 2938 publications, with the distribution as follows: PubMed contributed 2560 articles, DOAJ 157, and BioMed Central 221. After removing duplicates and further applying our inclusion and exclusion criteria (see Figure 1 for the criteria), 150 publications remained to be screened in more detail. These 150 publications were rechecked for inclusion criteria prior to extensive review; as shown in Figure 1, 42 articles were left to undergo extensive review.

### 2.2. Data Collection and Classification

Forty-two articles that met the inclusion criteria were reviewed, and a detailed summary table was constructed using Microsoft Excel to effectively organize the data for analysis (see Appendix A). For each publication, the following parameters were summarized: main author, year of publication, study objective, number of participants, mean age of participants, days of sample collection, sample type, methodology of data acquisition, statistical analysis techniques employed, identified metabolites, changes in metabolites levels (both upregulation and downregulation), metabolites linked to severity and/or mortality rates, and critical findings or conclusions drawn from the study. Additionally, a secondary table (Appendix A) was created to catalog the altered metabolites, facilitating the identification of recurring patterns in metabolite alterations across the reviewed studies. Furthermore, due to the identification of recurring patterns comparing mild and moderate/severe to critical COVID-19 cases, a tertiary table (see Appendix A) was created to aid in detecting metabolites that differ between these severity groups. Finally, two additional tables were created to summarize the number of upregulated and downregulated metabolites or metabolic ratios across studies (see Appendix A). Data were limited to human serum and plasma metabolites as these were the most frequently studied media. Pathway analysis was conducted.

Metabolites that appeared in three or more publications (recurrent identified metabolites involved in COVID-19) in the literature were compiled into a table (see Table 1) to compare mild COVID-19 cases with controls, as well as to compare data from moderate to severe cases of COVID-19 with controls (healthy or PCR-negative individuals). These recurrent metabolites were specifically chosen because the repeated findings strongly suggest the data are correct and that there was sufficient data to allow for a comparison between severity groups and controls. The identified recurrent metabolites were then analyzed to identify metabolic pathways associated with disease severity. NIH classification of disease severity was used wherever possible [5].

Pathway analysis was conducted using Metaboanalyst 6.0 [15]. A graph comparing pathway impact against statistical significance, expressed as −log_10_(p), was generated using the recurrent metabolites shown in Table 1 (see Figure 2a). A horizontal bar chart showing the statistical significance of each identified pathway was created using the Metaboanalyst findings and Microsoft tools (see Figure 2b).

## 3. Results

A total of 42 articles on COVID-19 metabolomics and its severity were analyzed in this scoping review. A summary table of the reviewed articles is shown in Appendix A. Appendix A summarizes the metabolites upregulated and downregulated in COVID-19 cases. In some articles, the cohort comparisons differ from the standard COVID-19-positive vs. COVID-19-negative (e.g., deceased vs. discharged, severe vs. non-severe, COVID-19 ARDS vs. non-COVID-19 ARDS); therefore, the nature of each comparison is specified in the respective columns. Additionally, a specific pattern was identified comparing the metabolic profiles of mild COVID-19 patients to those of moderate/severe COVID-19 patients. Consequently, a table summarizing the discriminatory metabolites between severity groups and the pathways involved was created based on 27 articles (Appendix A).

Two summary tables of recurrent upregulated and downregulated metabolites were created to understand better the relationship between the identified metabolites and COVID-19 severity (Appendix A). In this scoping review, a ‘recurrent metabolite’ is defined as a metabolite that appeared as increased or decreased in COVID-19 in three or more articles, indicating a more consistent and stronger discriminatory connection within severity groups. This information was then used to further compare the metabolic changes between severity groups and controls across studies (see Table 1).

### 3.1. Recurrent Metabolites Identified in Severity Groups Compared to Controls

Eighteen metabolites and one metabolic ratio were found to be recurrently changed and, in most cases, statistically significantly changed (*p* < 0.05) in plasma or serum samples when comparing severity groups to controls (healthy or negative PCR) in 11 studies. The metabolites that were decreased in mild COVID-19 cases compared to controls include leucine [16], phenylalanine [16], tyrosine [16], glucose [16], tryptophan [17,18], proline [19], glutamine [19], citrulline [20], citric acid [17], and isoleucine [20]. Conversely, metabolites that were increased in mild cases compared to controls were phenylalanine [17,19,21,22,23], lactate [23], glutamic acid [17,19,21,23], kynurenine [17], ornithine [23], xanthine [23], arachidonic acid [23], and 3-hydroxybutyric acid [22].

In moderate to severe COVID-19 cases compared to controls, the metabolites that showed decreased levels included tryptophan [17,18], proline [19], glutamine [19], citric acid [17,21], citrulline [20], and isoleucine [20]. In contrast, those with increased concentrations in this group were leucine [16], phenylalanine [16,17,19,22,23] , tyrosine[16], lactate [16,23], glucose [16,24], glutamic acid [17,19,21,23], kynurenine [17,18], C10:2 [17], ornithine [23], xanthine [23], arachidonic acid [23], 3-hydroxybutyric acid [21,22], and the kynurenine/tryptophan metabolic ratio [17,25].

### 3.2. Metabolites Increased or Decreased with Increasing Degree of Severity

Metabolites that increased with higher degrees of severity include leucine [16], phenylalanine [16,17,19,22], tyrosine [16], lactate [16], glucose [16,24], glutamic acid [17,19], kynurenine [17,18], and C10:2 [17], with phenylalanine, glucose, glutamic acid, kynurenine, and C10:2 being the most significant [16,17,18,19,24] (*p* < 0.001). Conversely, metabolites that decreased with progressing severity include tryptophan [17,18], proline [19], glutamine [19], citric acid [17,21], citrulline [20], and isoleucine [20], with tryptophan, proline, glutamine, citric acid, and citrulline being the most significant [17,19,20,21] (*p* < 0.001).

**Table 1 metabolites-14-00617-t001:** Recurrent metabolites across the literature allowed for comparison between severity groups and controls.

Metabolite	Mild Compared to Control: Change Identified	Moderate/Severe Compared to Control: Change Identified	Supplemental	Controls/Refs.
**Leucine**	Decreased	Increased	Levels increased with progressing severity. Plasma samples. ^1^	Healthy controls Correira et al. [16]
**Phenylalanine**	Decreased	Increased ***	Levels increased with progressing severity. Plasma samples. ^1^	Healthy controls Correira et al. [16]
	Increased *	Increased	Levels increased with progressing severity. Serum samples. ^1^	Healthy controls Paez-Franco et al. [19]
	Increased ***		Serum samples. ^1^	Negative PCR. Paez-Franco et al. [21]
	Increased ***	Increased ***	Levels increased with progressing severity. Plasma samples. ^2^	Negative PCR. Herrera-Van et al. [17]
	Increased *	Increased *	Levels increased with progressing severity. Serum samples. ^2^	Healthy controls Martinez-Gomez et al. [22]
	Increased *	Increased *	Serum samples. ^2^	Negative PCR Caterino M. et al. [23]
**Tyrosine**	Decreased	Increased **	Levels increased with progressing severity. Plasma samples. ^1^	Healthy controls Correira et al. [16]
**Lactate**		Increased **	Levels increased with progressing severity. Plasma samples. ^1^	Healthy controls Correira et al. [16]
	Increased *	Increased *	Serum samples. ^2^	Negative PCR Caterino M. et al. [23]
**Glucose**	Decreased	Increased **	Levels increased with progressing severity. Plasma samples. ^1^	Healthy controls Correira et al. [16]
		Increased ***	Levels increased with progressing severity. Plasma samples. ^2^	Negative PCR Villagrana- Bañuelos et al. [26]
**Tryptophan**	Decreased ***	Decreased ***	Levels decreased with progressing severity. Plasma samples. ^2^	Negative PCR Herrera-Van et al. [17]
	Decreased *	Decreased *	Levels decreased with progressing severity. Plasma samples. ^1^	Healthy controls Occelli C. et al. [18]
**Proline**	Decreased	Decreased ***	Levels decreased with progressing severity. Serum samples. ^1^	Healthy controls Paez-Franco et al. [19]
**Glutamic acid**	Increased **	Increased ***	Levels increased with the progressing severity. Serum samples. ^1^	Healthy controls Paez-Franco et al. [19]
	Increased ***	Increased ***	Higher concentration in mild compared to severe cases. Serum samples. ^1^	Negative PCR Paez-Franco et al. [21]
	Increased ***	Increased ***	Levels increased with progressing severity. Plasma samples. ^2^	Negative PCR Herrera-Van et al. [17]
	Increased *	Increased *	Serum samples. ^2^	Negative PCR Caterino M. et al. [23]
**Glutamine**	Decreased	Decreased ***	Levels decreased with progressing severity. Serum samples. ^1^	Healthy controls Paez-Franco et al. [19]
**Citric acid**		Decreased ***	Levels decreased with progressing severity. Serum samples. ^1^	Negative PCR Paez-Franco et al. [21]
	Decreased ***	Decreased ***	Levels decreased with progressing severity. Plasma samples. ^2^	Negative PCR Herrera-Van et al. [17]
**Kynurenine/Tryptophan**		Increased ***	Plasma simples. ^2^	Healthy controls D’Amora P et al. [25]
		Increased ***	Plasma simples. ^2^	Negative PCR
				Herrera-Van et al. [17]
**Kynurenine**	Increased ***	Increased ***	Levels increased with progressing severity. Plasma simples. ^2^	Negative PCR Herrera-Van et al. [17]
	Decreased *	Increased *	Levels increased with progressing severity. Plasma simples. ^1^	Healthy controls Occelli C. et al. [18]
**C10:2**		Increased ***	Levels increased with progressing severity. Plasma samples. ^2^	Negative PCR Herrera-Van et al. [17]
**Citrulline**	Decreased	Decreased	Levels decreased with progressing severity. Plasma samples.	Healthy controls Rahnavard A. et al. [20]
**Isoleucine**	Decreased	Decreased	Levels decreased with progressing severity. Plasma samples.	Healthy controls Rahnavard A. et al. [20]
**Ornithine**	Increased *	Increased *	Serum samples. ^2^	Negative PCR
				Caterino M et al. [23]
**Xanthine**	Increased *	Increased *	Serum samples. ^2^	Negative PCR Caterino M et al. [23]
**Arachidonic** **Acid**	Increased *	Increased *	Serum samples. ^2^	Negative PCR Caterino M et al. [23]
**3-hydroxybutyric acid**		Increased *	Serum samples. ^1^	Negative PCR Paez-Franco et al. [21]
	Increased *	Increased	Plasma samples. ^1^	Non-COVID Valdes A. et al. [22]

* *p* value < 0.05, ** *p* value < 0.01, *** *p* value < 0.001. ^1^ Untargeted metabolomics, ^2^ targeted metabolomics.

### 3.3. Pathway Analysis Related to COVID-19 Severity

Incorporating all recurrent metabolites listed in Table 1, we conducted a comparison between the severity groups with controls. The analysis identified the top pathways with the most significant impact on disease progression, which are as follows (Figure 2):

The impact score (Figure 2c) represents the relationship between the matched metabolites and the total number of metabolites within each pathway. A higher impact score indicates a more significant influence of changes in metabolite levels on that specific pathway [27].

Figure 2a illustrates the statistically significant pathways, as expressed by the −log(p) value. Notably, Figure 2b highlights the pathways with the highest statistical significance [−log(p)].

Pathway analysis reveals that arginine biosynthesis and phenylalanine, tyrosine, and tryptophan biosynthesis are the most significantly impacted metabolic pathways regarding impact score and statistical significance.

**Figure 2 metabolites-14-00617-f002:**
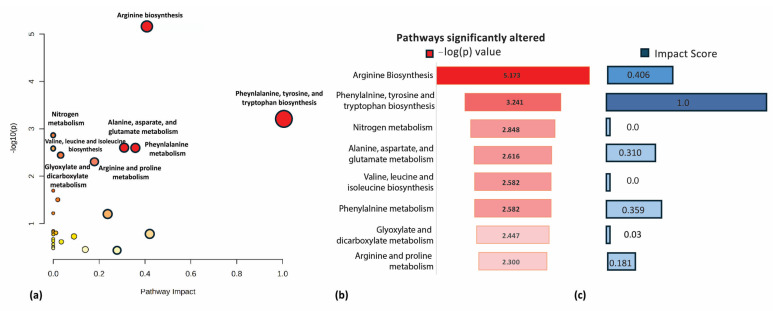
Pathway analysis of metabolites associated with COVID-19 severity: (**a**) An overview of pathway analysis indicating the significance -log(p) and pathway impact of recurrent metabolites identified across COVID-19 severity groups compared to controls. (**b**) A ranking of significantly altered metabolic pathways based on -log(p) values, highlighting the most affected pathways in the context of disease severity. These include: 1. Arginine biosynthesis; 2. Phenylalanine, tyrosine, and tryptophan biosynthesis; 3. Nitrogen metabolism; 4. Alanine, aspartate, and glutamate metabolism; 5. Valine, leucine, and isoleucine biosynthesis; 6. Phenylalanine metabolism; 7. Glyoxylate and dicarboxylate metabolism; 8. Arginine and proline metabolism. (**c**) The pathway impact score represents the relationship between the matched metabolites and the total number of metabolites within each of the involved pathways.

## 4. Discussion

Identifying recurrent metabolites in severity groups of COVID-19 compared to controls highlights the significant biochemical changes occurring in COVID-19 patients as identified in serum or plasma. The metabolites including phenylalanine, kynurenine, tyrosine, lactate, ornithine, glucose, citrate, citrulline, isoleucine, tryptophan, proline, glutamine, glutamate, xanthine, arachidonic acid, and 3-hydroxybutyric acid have been consistently reported to change in serum and/or plasma in patients with COVID-19 across multiple studies, reinforcing their relevance or correlation with the disease process. For instance, the elevated levels of phenylalanine, glucose, and kynurenine in severe cases suggest a disruption in amino acid metabolism and an increased inflammatory response, aligning with the production of proinflammatory cytokines (cytokine storm) observed in severe COVID-19 cases [20].

It is well established that hypoxemia in COVID-19 patients can impair energy production, immune responses, and redox balance, ultimately leading to systemic metabolic dysregulation [21]. In terms of energy metabolism, hypoxemia favors a change to anaerobic glycolysis over beta-oxidation, the pentose phosphate pathway, and oxidative phosphorylation as a source of ATP to meet the energy demands of critical processes such as RNA expression and protein and lipid synthesis [21]. Consequently, the increased glucose levels observed with greater disease severity correlate with elevated lactate levels due to anaerobic glycolysis or the less efficient use of aerobic glycolysis. Additionally, decreased levels of citrate, an intermediary of oxidative phosphorylation, correspond to increased COVID-19 severity or the presence of hypoxemia, which inhibits the influx of acetyl-CoA into the mitochondria, affecting citrate levels [21].

There is also evidence of immune dysregulation in COVID-19, with several recurrent metabolites described in the literature. These altered metabolites provide an insight into molecular changes in immune status found in subjects with COVID-19, including tryptophan, kynurenine, glutamine, citrulline, or ornithine. On the other hand, kynurenine itself—an indoleamine metabolite of tryptophan via the kynurenine pathway—has consistently been shown to be increased in COVID-19. Contrary to this, the levels of tryptophan decreased with COVID-19. Kynurenine has been positively correlated with increasing levels of proinflammatory cytokines and C-reactive protein [20]. Furthermore, kynurenine has been shown to modulate T-cell differentiation, promote endothelial activation, and decrease microvascular reactivity [20].

Further, tryptophan and glutamine have been reported to support immune cell proliferation [28]. Notably, low tryptophan levels have become a strong marker of fatality in COVID-19 patients [28], whereas glutamine deprivation may further hamper M2 macrophage polarization. M2 macrophages belong to the subset of cells that play a crucial role in resolving inflammation and promoting tissue repair; supporting immune regulation [29]. Although glutamine nutrition supplementation in critically ill patients, including those with COVID-19, has been associated with shorter hospitalization periods, the interesting fact is that it has not really been explored yet. Additionally, both citrulline and ornithine may be metabolized to furnish arginine and potentially also contribute to immunity against SARS-CoV-2 infection [29].

According to Paez-Franco et al. [19] decreased oxygen supply, enhanced glycolysis, and the activation of immune responses enhance an altered redox state in COVID-19. The synthesis, recycling, and supplementation of antioxidants like glutathione GSH and ascorbic acid are vital for clearing ROS [19]. Glutamine, GSH, and sometimes proline metabolism can directly influence the amount of glutamate and cystine—both amino acids involved in GSH synthesis—in the body [19]. Lower levels of glutamine and proline with higher levels of glutamate, associated with increasing COVID-19 severity, could generate immune dysfunction in addition to a reduced antioxidant capacity in seriously ill patients. Along this line of thinking, the therapeutic use of *N*-acetyl-cysteine is also associated with a reduced risk of mechanical ventilation and mortality in patients with COVID-19 by increasing the body’s antioxidant capacity [19].

SARS-CoV-2 infection can cause serious issues with organ dysfunction and overall body or organ dysregulation. Researchers have pinpointed certain metabolites that offer clues about these complications. Recurrent metabolites like citrulline, isoleucine, ornithine, and phenylalanine may indicate organ dysfunction or systemic issues. For instance, low citrulline levels in both mild and severe COVID-19 cases are connected to gastrointestinal symptoms and systemic inflammation [20]. Since citrulline is made by enterocytes in the small intestine, low levels might suggest a reduction in enterocyte function [20]. These cells also have ACE2 receptors, which the virus targets, making it likely that gut issues and low citrulline levels in COVID-19 patients are associated with cell damage [20]. Ornithine, another important metabolite in the urea cycle, is associated with liver dysfunction, especially when the citrulline/ornithine ratios are low in severe cases of COVID-19 [25]. Isoleucine, a branched-chain amino acid (BCAA) that is with other BCAAs crucial for protein building, tends to be low in conditions like liver cirrhosis, urea cycle disorders, and chronic kidney failure. Its decrease in severe COVID-19 cases likely reflects kidney dysfunction [20]. The decrease in isoleucine levels with increasing COVID-19 severity, as identified in this scoping review, likely correlates with the impaired renal function seen in patients with severe COVID-19 [20]. Phenylalanine is another metabolite often linked to systemic problems. High phenylalanine levels can result from rapid protein breakdown [17] and are associated with worse outcomes in COVID-19 patients [28]. Elevated levels of phenylalanine are also associated with microvascular endothelial damage and higher coagulation risks, which are common in severe COVID-19 patients. Elevated levels of phenylalanine are also seen in patients with COVID-19-induced heart disease [29].

Other metabolites such as arachidonic acid (AA) participate in the production of inflammatory mediators, including prostaglandins and leukotrienes, which may contribute to the cytokine storm and inflammation seen in severe cases of COVID-19 and could aggravate lung injury with systemic inflammation [30]. AA itself could be an endogenous antiviral; its deficiency may, therefore, make one vulnerable to COVID-19 [31]. Therapeutics targeted at inhibiting one or more enzymes in the AA pathway, such as COX-2, could help modulate the severity of COVID-19 by regulating excessive inflammation. These findings could, therefore, link plasma levels of AA and its metabolites to the severity of COVID-19 and potentially serve as disease progression biomarkers [32]. Decanoic acid (C10:1), a mono-unsaturated form of C10, is somewhat lacking in available information about its potential role in COVID-19 and disease severity. However, capric acid, together with other medium- and short-chain fatty acids, may downregulate the proinflammatory cytokines TNFα, IL-6, and IL-12, contributing to potentially reducing inflammation and preventing the development of COVID-19 symptoms and also inducing anti-inflammatory cytokines like the aforementioned immunosuppressive cytokine, IL-10 [33]. It is naturally inherent in fatty acids to exert a specific antimicrobial effect to disrupt viral, fungal, and bacterial cell membranes in a modulatory way on the virus infection, including affecting replication. However, COVID-19 might have a large effect on the modulation of lipid metabolism, thus including capric acid and other fatty acids whose profile alterations were related to the severity of this disease; thus, it is important to consider how FAs may modulate the course and outcome of COVID-19.

### 4.1. Pathway Analysis and Implications

The pathway analysis further reinforces the significance of these metabolic changes, particularly in the context of amino acid biosynthesis and nitrogen metabolism. The high impact scores observed based on the changes in metabolites with COVID-19 were found in the following pathways: phenylalanine, tyrosine, and tryptophan biosynthesis, as well as arginine biosynthesis, indicating that disruptions in these pathways may be key drivers of COVID-19 severity. The statistical significance of arginine biosynthesis, highlighted by its -log(p) value, suggests that arginine metabolism may be a critical pathway involved in modulating the immune responses, and it also is involved in nitric oxide production, both of which are essential in the body’s defense against viral infections [34].

### 4.2. Limitations

Although standardized severity scales for defining COVID-19 severity, such as those proposed by the World Health Organization (WHO) [35] and the National Institutes of Health (NIH) [4], are available, not all metabolomic studies on COVID-19 severity have used these scales. This inconsistency may be because these scales were not yet developed or finalized when some studies were conducted. As a result, there is a lack of uniform categorization of COVID-19 severity across studies, with many classifying severity based on criteria such as ICU admission, oxygen requirements, PaO_2_/FiO_2_ ratio, or Berlin’s definition for ARDS, among others. Additionally, the variability in study design has led to heterogeneity of findings across studies, complicating the interpretations of the results. There is a lack of standardized metabolomics approaches, which contributes to inconsistent identification and quantification of metabolites across studies. Furthermore, the interpretation of metabolic pathways may not fully capture the complexity of metabolic changes associated with COVID-19. In addition, the focus on serum and plasma samples in many studies may not represent the metabolic changes occurring in other tissues or organs, limiting the comprehensive understanding of the disease’s metabolic impact. Finally, although this study identified significantly altered metabolites associated with varying degrees of COVID-19 severity, there is a lack of validation in any of these preliminary studies to reveal true biomarkers for diagnosing COVID-19 severity and guiding clinical treatment. The complex nature of COVID-19 infection—including factors such as different viral variants, treatments, vaccination status, pre-existing comorbidities, and other confounding variables not escribed in many of these studies—highlights the need for more rigorous research to establish these findings.

## 5. Conclusions

Based on the comprehensive data presented in this scoping review, we conclude that COVID-19 severity is intricately linked to significant metabolic alterations that span amino acid metabolism, energy production pathways, immune response modulation, and redox balance. Recurrent metabolites outlined in this scoping review have been consistently identified across multiple studies, indicating their potential as biomarkers for assessing disease severity. The metabolic shifts observed in severe COVID-19 cases, including increased kynurenine and phenylalanine levels, decreased glutamine and tryptophan, and altered glycolysis and oxidative phosphorylation, underscore the profound biochemical changes accompanying the disease and its severity. Furthermore, the correlation between these metabolic changes and clinical outcomes, such as immune dysregulation, organ dysfunction, and systemic inflammation, highlight the potentially useful role of metabolomics in understanding the pathophysiology of COVID-19.

Despite these findings, the variability in study designs, metabolomics methodology, severity categorizations, and external factors such as treatment protocols and patient demographics suggest the necessity of further studies. Establishing a standardized approach to categorize COVID-19 severity, metabolomics evaluation and a more extensive exploration of confounding variables will enhance the reliability of these metabolic markers as diagnostic tools and therapeutic targets. Future research should focus on validating these findings to fully harness the potential of metabolomics in managing COVID-19. Through such efforts, we can better understand the molecular mechanisms driving the disease progression and improve patient outcomes by potentially tailoring therapeutic strategies to individual metabolic profiles.

## Figures and Tables

**Figure 1 metabolites-14-00617-f001:**
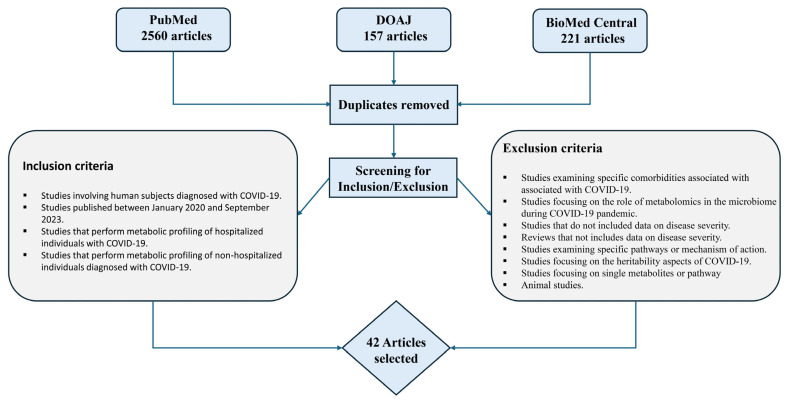
Summary of searched databases, inclusion, and exclusion for review using PRISMA criteria.

## Data Availability

The original contributions presented in this study are included in the article/Appendix A. Further inquiries can be directed to the corresponding author.

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
