# Peer review of "Metabolomic Insights into COVID-19 Severity: A Scoping Review"

_metabolites, 2024, doi:10.3390/metabo14110617_

Round 1
Reviewer 1 Report
Comments and Suggestions for Authors
Manuscript Number: metabolites-3258774-peer-review-v1
Title:"Metabolomic Insights into COVID-19 Severity: A scoping review”
Review Report
The manuscript titled "Metabolomic Insights into COVID-19 Severity: A scoping review”by Pimentelet al.addresses a very important and relevant topic that aims to decipher deeper insights int how the metabolic changes are affected with the severity of COVID-19, based on published reports from 2020-2024. Using a systematic approach and PRISMA guidelines, authors retrieved the relevant publications from three databases using precise keywords. Using defined inclusion exclusion criteria, 42 articles reporting metabolomic studies on sampled with defined level of disease severity. Comparative analysis of these reports concluded in the identification of key metabolites, which increases or decreases with the increase in the severity of the disease. While phenylalanine, glucose and glutamic acid increases with severity, tryptophan, proline and glutamine decreases. Further, pathway analysis revealed that biosynthetic pathways of phenylalanine, tyrosine and tryptophan were most significantly affected as well as arginine biosynthetic pathway.
The work is well designed, very well written and easy to follow. The study and the workflow are meticulously designed, executed. The findings are well described and the conclusions are clearly drawn. The findings are clearly discussed in the backdrop of current literature. Further, discussion on pathway analysis yield clear insights into how the observed metabolic changes are related to immune dysregulation and organ dysfunction.
Overall, this paper is a valuable contribution to the field, offering a thorough review of how metabolomic changes correlate with COVID-19 severity.
Minor comments/corrections:
1. Page 3, Figure 1.PubMed 2,500 articles; whereas in the text, precise number is given (line 110: PubMed contributed 2560 articles). It’s better to use precise number of articles in figure 1 as well.
2. The entire paper is very well written, however, the language in the Introduction section could simplified to some extent for better understanding of the reader.
Author Response
Title:"Metabolomic Insights into COVID-19 Severity: A scoping review”
"We appreciate the reviewers’ comments and suggestions. We have revised the manuscript accordingly as highlighted below.
Response to Reviewer #1
Minor comments/corrections:
- Page 3, Figure 1.PubMed 2,500 articles; whereas in the text, precise number is given (line 110: PubMed contributed 2560 articles). It’s better to use precise number of articles in figure 1 as well.
Response: Thank you for your comment. we updated the figure 1 with correct number 2560.
- The entire paper is very well written, however, the language in the Introduction section could simplified to some extent for better understanding of the reader.
Response: Thank you for your comment. We had made some change the introduction to helpfully make it more understandable.
Reviewer 2 Report
Comments and Suggestions for Authors
This review is an analysis of published studies on changes in the blood metabolome of people who have had SARS-Cov-2 infection. The aim of this work was to identify changes in the metabolome corresponding to different degrees of disease severity. The authors of the manuscript have done a great job; they analyzed a really large array of data. The methods and approaches that were used to analyze the data are adequate. There is no doubt about the reliability of the data obtained. The result of the work was a table listing metabolites - potential markers of the disease. Undoubtedly, this is useful information for subsequent studies. The conclusions correspond to the results of the analysis.
The article is written in a clear, concise style. In terms of fundamental significance, the article can definitely be published. I have only a few comments:
Line 136-138: What did the fifth author do? I think this text can be omitted.
Line 199-200: Repeat. This was already said earlier in Line 149-150.
Line 203: Data missing from the figure.
Line 214-218: Unnecessary, since the reader can easily read this information from the figure.
A few questions about supplementary table 1:
What does the red lines mean? Some literature sources are marked with an asterisk, what does this mean?
Some articles have empty cells (year, author, purpose), some have all empty cells. Please, clarify.
Author Response
Title:"Metabolomic Insights into COVID-19 Severity: A scoping review”
"We appreciate the reviewers’ comments and suggestions. We have revised the manuscript accordingly as highlighted below.
Response to Reviewer #2
Line 136-138: What did the fifth author do? I think this text can be omitted.
Response: Thank you for your comment. We have explained this to clarify it as we believe it is an important piece of information for the reader.
Line 199-200: Repeat. This was already said earlier in Line 149-150.
Response: Thank you, we remove it.
Line 203: Data missing from the figure.
Response: Thank you, we integrated it into figure 3c and removed it from the text
Line 214-218: Unnecessary, since the reader can easily read this information from the figure.
Response: Thank you, we removed it.
A few questions about supplementary table 1:
What does the red lines mean? Some literature sources are marked with an asterisk, what does this mean?
Response: Thank you for your attention, we removed the asterisk and mentioned the highlighted cell in green (sheet 3) as selected papers to the study.
Some articles have empty cells (year, author, purpose), some have all empty cells. Please, clarify.
Response: Thank you, we have refilled the empty cells and apologize for missing this in our review of the material before submission.